# ZERO-SHOT DETECTION OF DAILY OBJECTS IN YCB VIDEO DATASET

## ABSTRACT

To let robots be able to manipulate objects, they have to sense the location of objects. With the development of visual data collecting and processing technology, robots are gradually evolving to localize objects in a greater field of view rather than being limited to a small space where the object could appear. To train such a robot vision system, pictures of all the objects need to be taken under various orientations and illumination. In the traditional manufacturing environment, this is applicable since objects involved in the production process does not change frequently. However, in the vision of smart manufacturing and high-mix-low-volume production, parts and products for robots to handle may change frequently. Thus, it is unrealistic to re-training the vision system for new products and tasks. Under this situation, we discovered the necessity to introduce a hot concept which is zero-shot object detection. Zero-shot object detection is a subset of unsupervised learning, and it aims to detect novel objects in the image with the knowledge learned from and only from seen objects. With zero-shot object detection algorithm, time can be greatly saved from collecting training data and training the vision system. Previous works focus on detecting objects in outdoor scenes, such as bikes, car, people, and dogs. The detection of daily objects is actually more challenging since the knowledge can be learned from each object is very limited. In this work, we explore the zero-shot detection of daily objects in indoor scenes since the objects' size and environment are closely related to the manufacturing setup. The YCB Video Dataset is used in this work, which contains 21 objects in various categories. To the best of our knowledge, no previous work has explored zero-shot detection in this object size level and on this dataset.

## 1 INTRODUCTION

Industrial robots have received more and more attention in the manufacturing industry due to the rising cost of human labour and decreasing cost of industrial robots (Carlisle, 2017). Since robots can handle heavy and repetitive jobs better than human, many manufacturing planets have replaced human labours on the production line with robots (Robla-Gómez et al., 2017). In today's manufacturing pattern of mass production, an industrial robot is only in charge of a certain processing step with dedicated parts. This manufacturing scenario does not require the robot to change target objects to work with frequently. However, with the recent development of control and communication technologies, the manufacturing industry is gradually evolving to high-mix-low-volume production that provides personalized product for customers (Lu et al., 2020). This is also known as smart manufacturing, in this scenario, the manufacturing system will become more flexible. Instead of tied to a specific task, robots will be allocated to various tasks depend on demand. Which requires the robots to be able to recognize a wide range of objects that could be involved during production.

With the development over years, today's object detection and recognition algorithms such as Faster-RCNN (Ren et al., 2015), SSD (Liu et al., 2016), YOLO (Redmon et al., 2016) and EfficientDet(Tan et al., 2020) have reached a high performance. With enough collected training data, those off-the-shelf algorithms can be easily applied to the robots' vision system to recognize and localize objects involved in the production process. However, the data collection, labelling and training of a neural network is a time-consuming process and requires expertise in the machine vision filed. Even there are data generation method that can generate synthetic images and labels from CAD models

(Wohlhart & Lepetit, 2015), training a new neural network for additional new parts frequently is still not realistic in personalized production.

Zero-shot learning (ZSL) is a learning paradigm that learns knowledge from seen categories and apply the knowledge to new categories in order to recognize objects that is never seen before. In the work carried out by Zhang & Saligrama (2015) and Zhang & Saligrama (2016), zero-shot learning algorithms have already achieved a reasonable accuracy on classifying unseen objects. While zero-shot learning only aim to recognize unseen categories, a more realistic problem called generalized zero-shot detection (gZSL) is proposed by Xian et al. (2017), which aims to recognize both seen and unseen categories. However, gZSL still have flaws and cannot be directly applied to solve the previously mentioned issues. For both ZSL and gZSL, they only focus on recognizing the object in an image. Thus, a big assumption is made before applying those algorithms, which is only one object is appearing in the image and it is always located in the middle of the image. In this setting, ZSL and gZSL algorithms only need to analysis the categorical information in the image but not the location information. In other word, it takes the whole image as one object proposal. In real life, cameras attached on robots are always moving with the robot and cannot make sure the target object is always located in the middle of the camera's view. Thus, generalized zero-shot detection (gZSD) and gZSL have to be combined to achieve a vision system that can detect and recognize unseen objects. Where gZSD oversees localize seen and unseen objects in the field of view and gZSL oversees the recognition and categorization of seen and unseen objects.

They are existing works in this field try to solve gZSD and gZSL at the same time to make the whole scenario completed. For example, Rahman et al. (2018) tried to combine Faster RCNN with gZSL module to generate object proposals and class predictions. Zhu et al. (2019) worked on YOLOv2, using a single stage detector to generate object proposals. Previous works have all worked on outdoor datasets such as MS COCO (Lin et al., 2014) and Pascal VOC(Everingham et al., 2010). Different objects in those datasets are categorized in categories. For example, man, woman and children are very different in term of their visual appearance but they are all allocated into the 'people' category. The varieties of objects in one category can increase the generality of the trained algorithm. In the case of detecting daily objects, datasets such as YCB Video (Xiang et al., 2017) is chosen. 3D objects in this dataset is similar to the size and characteristics of the objects could appear in production pipeline. However, each object is unique and cannot be categorized into categories in 3D object datasets. Which brings the challenge of harder generalization to unseen objects.

Regarding the problems and challenges we found in future manufacturing environment and gZSD, we propose to modify the base version of YOLOv5 (Jocher et al., 2021) to perform gZSL on the YCB Video dataset. Compare to two stage detectors such as Faster RCNN, one stage detectors such as YOLO and SSD are much faster to process. YOLOv5 as the latest version of YOLO series detectors, has been proved to be faster and better than previous versions. Compare to the work done by Zhu et al. (2019) which used a modified YOLOv2 (Redmon & Farhadi, 2017) to perform gZSD, YOLOv5 can output objects proposals in three different levels, thus have better coverage on object sizes. For training and testing our algorithm, four objects out of 21 objects in YCB Video dataset are picked as unseen objects. Any image that contains these four unseen objects will never appear during training but will be used during testing. For every object, their class labeling is translated into attribute vectors that's contains the colour and shape information of each object. Thus, we transformed the classical single label problem into a multi-label problem to let the neural network learn attribute labels and apply it to unseen objects. It needs to be notice that, in this work, we only work on the gZSD problem but not gZSL. Which mean we only aim to localize seen and unseen object in the images by bounding boxes but not define the class label of the objects.

Our contributions in this paper are in three folds: 1.A novel neural network structure that based on YOLOv5 and able to perform generalized zero-shot detection. The output bounding boxes can be further combined with other gZSL algorithm to achieve full zero-shot object detection and recognition. 2.A novel splitting method for YCB Video dataset that splits the dataset by seen and unseen objects. This splitting can be used for both gZSD and gZSL research that related to daily objects. 3.A novel attribute labelling method for objects in YCB Video dataset. Covert the class labelling to 16 attributes that represents colour and shape information of an object for the neural network to learn.

## 2 RELATED WORK

### 2.1 OBJECT DETECTION

Researches on object detection and recognition have been developing rapidly in the past decade. The earliest image classification algorithm can be traced to the work published by Krizhevsky et al. (2012). Since then, the recognition speed and accuracy of image classification algorithms have been improving continuously. These algorithms can be divided into two categories: two-stage detectors and one-stage detectors. Two-stage detectors such as Faster RCNN Ren et al. (2015)and R-FCN Dai et al. (2016)generate object proposals by Region Proposal Network (RPN), then perform object classification based on these proposals. One-stage detectors like (Liu et al., 2016), YOLO (Redmon et al., 2016) and EfficientDet(Tan et al., 2020) generate object proposals and classify objects at the same time by dividing the image into grids. Thus, one-stage detectors' processing speed is faster than two-stage detectors' and gained more attention recently.

YOLOv5 (Jocher et al., 2021) used in this paper is the fifth version of the classical YOLO detector. However, there is still an argument in this field about whether this algorithm is qualified to be considered as the fifth version. The maintainer of YOLOv5 also has not published a paper to justify the algorithm's ability. However, it is tested that on the MS COCO dataset (Lin et al., 2014), YOLOv5's speed and accuracy both outperformed the state-of-the-art algorithm, which is Google's EfficientDet. YOLOv5 is composed of three parts: Backbone, Neck and Head. When an image is passed into the network, it is first processed by the DarkNet (Bochkovskiy et al., 2020) backbone. Then passed into the PANet (Wang et al., 2019) neck, which processes and split the feature map into three different feature levels to have better coverage on objects in different sizes. Finally, the YOLO detector head will output predictions in three levels base on the feature maps. In this work, we will use YOLOv5 as the base algorithm, modify the detectors in the head part of the neural network, enable it to detect both seen and unseen objects.

### 2.2 ZERO-SHOT LEARNING

Given images with class labels, zero-shot learning (ZSL) aims to classify an unseen class based on knowledge learned from seen classes (Fu et al., 2018). ZSL works by learning semantic information from seen classes and reassemble the semantic attributes to predict unseen classes (Zhang & Saligrama, 2016)(Rahman et al., 2018). In other words, the algorithm learns the mapping from the visual domain to the semantic domain during learning and make predictions by mapping from the semantic domain to the visual domain. However, ZSL algorithms are designed for recognizing unseen classes, their performance degrades when both seen and unseen classes need to be recognized.Xian et al. (2017) proposed generalized ZSL (gZSL), which released the constraint of recognition targets and included seen classes as well. However, ZSL and gZSL both assume that only one target exists in an image and it is located right in the middle of the image. They still lack the ability to isolate an object from the background or the occlusion of other objects.

### 2.3 ZERO-SHOT DETECTION AND RECOGNITION

There are several methods proposed recently to solve the zero-shot detection and recognition problem. These algorithms can not only detect the location of seen and unseen objects in an image but can also classify them. Most of them took the approach with two-stage detectors. In the research carried out by Bansal et al. (2018), Edge-Box was used to generate regional proposals, and Regional Proposal Network (RPN) was used in work done by Rahman et al. (2018). Using two-stage detectors is an easier way to achieve generalized zero-shot detection (gZSL) since these proposal generators do not need to be trained to work with unseen objects. They will generate proposals regardless of the content inside the bounding box. The confidence and class prediction are handled by following gZSL network. However, these methods are inevitably slow due to the workload of class evaluation that comes with a large number of regional proposals. Zhu et al. (2019) proposed to use YOLOv2 as the backbone to detect and classify unseen objects. The use of one-stage detector made an improvement in terms of detection speed compared to two-stage detectors.

The mentioned methods above have all focused on outdoor scenes, and their commonly used datasets are MS COCO (Lin et al., 2014) and Pascal VOC (Everingham et al., 2010). The objects inside these

two datasets have a high variability within each class. For a robot that works in an indoor environment, recognizing objects at class level is not good enough since objects belong to the same category may have very different usage. For example, a housekeeping robot should be able to differentiate the blue cup and the pink cup and hand them to a boy and a girl, respectively, rather than recognize them both as cups. Abdalwhab & Liu (2019) have tried to use SUN RGB-D dataset (Song et al., 2015) to perform gZSD in an indoor environment. However, objects in the SUN RGB-D dataset is labelled by class and in furniture size. In this work, we will use the YCB Video dataset (Xiang et al., 2017), which includes 21 distinctive objects in desktop size. The object size and environment in YCB Video dataset are closer related to the setup we may encounter in the manufacturing environment.

## 3 METHODOLOGY

### 3.1 PROBLEM DEFINITION

Assuming we have $n$ objects labelled as $\boldsymbol{g}_i = \{\boldsymbol{b}_i, \boldsymbol{a}_i\}_i^n$. Where the 4-dimensional vector $\boldsymbol{b}_i = \{x_i, y_i, w_i, h_i\}$ denotes the ground truth bounding boxes'center point location $x, y$, width and height $w, h$. $\boldsymbol{a}_i$ is a 16-dimensional vector describes the feature attribute of the object $\boldsymbol{g}_i$. In the training dataset, all objects are all seen objects, represented by $\boldsymbol{g}_i \in \mathbb{O}_{seen}$. The valadation dataset is also composed by seen objects only. In the test dataset, both seen $\mathbb{O}_{seen}$ objects and unseen objects $\mathbb{O}_{unseen}$ are present, $\mathbb{O}_{seen} \cap \mathbb{O}_{unseen} = \emptyset$. The goal of this work is to predict $\{\boldsymbol{b}_{pred}, \boldsymbol{c}_{pred}, \boldsymbol{a}_{pred}\}$ for $\boldsymbol{g}_{pred} \in \mathbb{O}_{seen} \cup \mathbb{O}_{unseen}$. The extra term $\boldsymbol{c}_{pred}$ represents the confidence level of the existance of an object within bounding box $\boldsymbol{b}_{pred}$.

### 3.2 NETWORK ARCHITECTURE

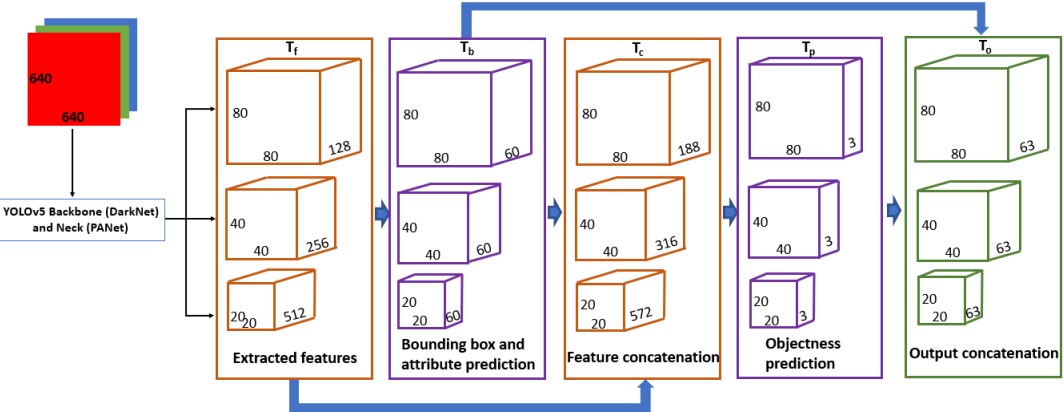

Figure 1: Simplified structure of our YOLOv5-ZS network

The simplified architecture of our YOLOv5-ZS network is shown in Figure 1. The network takes an RGB image as input and output predictions in three feature levels. The input image size is 640*640*3 in our network since all images in the YCB Video dataset are 640*480 pixels. They are padded with grey pixels on top and bottom to become a square shape. The image is first processed by the Backbone (DarkNet) and Neck (PANet). The size of DarkNet and PANet is changeable in YOLOv5. By changing the number of convolutional layers and feature map depth, four versions of YOLOv5can be created, and they are YOLOv5s (small), YOLOv5m (medium), YOLOv5l (large) and YOLOv5x (extra-large). As the network becomes bigger and deeper, their detection accuracy will increase, but the detection speed decreases. In our work, we chose to use YOLOv5s. Tensor $\mathbf{T}_f$ represents the block of are extracted features in three levels, they are passed to the following blocks: bounding box and attribute prediction ($\mathbf{T}_b$), feature concatenation ($\mathbf{T}_c$), objectness prediction ($\mathbf{T}_p$) and output concatenation ($\mathbf{T}_o$). They are detailly explained in the following sections.

### 3.2.1 FEATURE EXTRACTION

The $\mathbf{T}_f$ block is composed of three tensors, and its structure is inherited from PANet. Path Aggregation Network (PANet) proposed by Wang et al. (2019) is a network that has both top-down and bottom-up dataflow. The bi-directional data flow ensures the image features are kept better in the later convolutional layers. By using PANet, we can also get image features in three levels. As Figure 1 shows, the feature tensors are in size 80*80*128, 40*40*256, 20*20*512. As the network goes deeper, the feature map gets smaller and deeper. From this block, each data block will consist of three tensors. However, for easier representation and clearer graph, we represent each later block as a tensor $\mathbf{T}$.

### 3.2.2 OBJECT LOCALIZATION

The location of bounding boxes is predicted in block $\mathbf{T}_b$. Each tensor in this block has the same depth of 60. In YOLO series detectors, the width and height of the detection layer represent the number of grids on the image. For example, 80*80 means the image is evenly divided into 6400 grids, and each grid is responsible for predicting bounding boxes that the center point falls on this grid. As the grid number becomes smaller, the size of each grid becomes bigger and hence have a better focus on bigger objects. The three different grid sizes thus allow the network to detect objects in various sizes. Each tensor's depth in this block is 60 since the network needs to make three predictions with anchors in different width/height ratios for each grid. For each prediction, $\boldsymbol{b}_p$ is 4 digits and $\boldsymbol{a}_{pred}$ is 16 digits. Thus the tensor has 3*(4+16) = 60 channels. It needs to be notice that the bounding box prediction $\boldsymbol{b}_p = \{t_x, t_y, t_w, t_h\}$ are relative to the location of the grid and size of the anchor. The actual location and size of the bounding box $\boldsymbol{b}_{pred}$ need to be calculated with the following equations.

$$x = 2 * \sigma(t_x) - 0.5 + c_x \qquad\qquad y = 2 * \sigma(t_y) - 0.5 + c_y$$
$$w = p_w * (2 * \sigma(t_w))^2 \qquad\qquad h = p_h * (2 * \sigma(t_h))^2$$

Where $c_x, c_y$ indicate the location of the top-left corner of a grid cell and $p_w, p_h$ indicate the width and height of the anchor.

### 3.2.3 ATTRIBUTE PREDICTION

As mentioned in the previous section, block $\mathbf{T}_b$ also needs to predict the 16-dimensional attribute vector for each object. Unlike other works describing objects with semantic vectors that learn by Word2Vec or FastText, our attribute vector is carried out by human eye evaluation. The elements in the attribute vector are some common colors and shapes that appear in all objects. There are two reasons that we took a different approach. **(1)** Class names such as "people" in previous works can be easily translated into semantic vectors using existing algorithms. While object names in YCB Video dataset are instance-specific, such as "master chef can", cannot be directly translated. **(2)** There is no visual variation for each object, and we can determine the attributes an object contains by human evaluation. The 16-dimensional attribute vector contains: white, blue, red, yellow, silver, black, brown, bottle, cup, can, clamp, slim, circle, cylinder, box, rectangular. Each object $\boldsymbol{g}_i$ will be described by several attributes in the form of one-hot embedding. For example, object 'red cup' has the attributes of red, cup, circle, cylinder will have the attribute vector [0 0 1 0 0 0 0 0 1 0 0 0 0 1 1 0 0]. The corresponding location of red, cup, circle, cylinder is labelled '1' and the rest are labelled '0'. The predicted output $\boldsymbol{a}_{pred}$ is a 16-dimensional vector of floating-point numbers. Each number's value is between 0 and 1, indicate the confidence level of an attribute.

### 3.2.4 CONFIDENCE PREDICTION

After bounding box and attribute prediction, block $\mathbf{T}_f$ and block $\mathbf{T}_b$ are concatenated together to form block $\mathbf{T}_c$. Our objectness prediction layer is learned from the concatenated layer $\mathbf{T}_c$. In the original YOLOv5 detection layer, objectness confidence is learned in the same block as $\mathbf{T}_b$. However, learning objectness confidence only from the feature layer will cause the network to only recognize seen objects and treat all unseen objects as background. Thus, we concatenate the $\mathbf{T}_f$ block and $\mathbf{T}_b$ block together to let the network also learns from the bounding box and attribute predictions. In this case, the network will be able to recognize unseen objects by the attributes they have. Detectors in $\mathbf{T}_p$ only have three channels, each channel of a grid cell is the confidence score

of the corresponding bounding box. The network will make (80*80+40*40+20*20) *3 = 25200 predictions in total. In the end, the output block $\mathbf{T}_o$ is the concatenation of bounding box, attributes and objectness prediction.

## 3.3 LOSS FUNCTION DESIGN

The total loss in our algorithm is composed of three parts: localization loss, attribute loss and objectness loss. In the following sections, we will show how the loss functions are designed and implemented.

### 3.3.1 LOCALIZATION LOSS

In YOLOV5, the localization loss is calculated by cIoU loss proposed by Zheng et al. (2021). Compare to the original IoU loss, cIoU loss is more precise and converges much faster. To calculate the cIoU loss, we need to calculate some parameters first:

$$IoU = \frac{Area_{pred} \cap Area_{gt}}{Area_{pred} \cup Area_{gt}}$$

$$\alpha = \frac{v}{(1 - IoU) + v}$$

$$v = \frac{4}{\pi^2} * (arctan\frac{w_{gt}}{h_{gt}} - arctan\frac{w_{pred}}{h_{pred}})^2$$

When the loss is calculated, not all bounding box predictions are used. A predicted bounding box is used only when its center point falls into the same grid cell as the ground truth bounding box's center point and has the highest IoU among three predictions in the same grid cell. We denote the selection of bounding box predictions using $\lambda_i$, it is set to 1 when selected otherwise 0. Since our output has three different feature levels, we define $n$ as the level number. The total number of predictions $m$ is equal to 80*80*3=19200, 40*40*3=4800, 20*20*3=1200 when $n$ is equal to 1, 2, 3 respectively. The formal localization loss is defined as the summation of mean cIoU loss in each layer, shown in the following function. Where $d$ is the distance between two boxes' center and $c$ is the diagonal length of the minimum enclosing box of two boxes.

$$L_{loc} = \sum_{j=1}^{n}(\frac{1}{m}\sum_{i=1}^{m}\lambda_i(1 - IoU_i + \frac{d_i^2}{c_i^2} + \alpha_i v_i))$$

### 3.3.2 ATTRIBUTE LOSS

For calculating the attribute loss, we used Binary Cross Entropy (BCE) loss with sigmoid function ($\sigma$). Since the ground-truth value of an attribute $e^{gt}$ is either 0 or 1, the predicted attribute value $e^{pred}$ need to be passed into a sigmoid function first to regulate the number to between 0 and 1. A new term $z$ is introduced in this function, and it represents the total number of attributes in the vector, which is 16. A bounding box's attribute loss is the summation of BCE loss on every attribute term. All other symbols remain the same meaning as in section 3.3.1.

$$L_{att} = \sum_{j=1}^{n}(\frac{1}{m}\sum_{i=1}^{m}\sum_{k=1}^{z}\lambda_i(e_{i,k}^{gt}(-\log(\sigma(e_{i,k}^{pred})) + (1 - e_{i,k}^{gt})(-\log(1 - \sigma(e_{i,k}^{pred})))))$$

### 3.3.3 OBJECTNESS LOSS

Different from localization loss and attribute loss, the objectness loss is calculated from all predictions rather than positive predictions only. Thus, the term $\lambda_i$ is dropped in the objectness calculation. BCE loss with sigmoid function is also used in the objectness loss calculation. $p^{gt}$ is the ground truth probability of the presence of an object in the bounding box, which equals 1 when an object is present and 0 otherwise. $p^{pred}$ is the predicted confidence score, regulated between 0 and 1 with the sigmoid function.

$$L_{obj} = \sum_{j=1}^{n} (\frac{1}{m} \sum_{i=1}^{m} p_i^{gt} (-\log(\sigma(p_i^{pred}))) + (1 - p_i^{gt})(-\log(1 - \sigma(p_i^{pred}))))$$

## 4 EXPERIMENTS

### 4.1 DATASET SETTING

In the following Table 1 we show how the original YCB Video dataset is splitted to our train, validation and test dataset. The YCB Video dataset is consist of 92 videos, and each has thousands of frames. 21 daily objects are included in the dataset, and some of them are placed in the scene of a video. Since during the video taking, the setup of objects does not change, objects in a video record remain constant. Thus, once the unseen objects is picked from all objects, all images that contain any of these four objects need to be allocated to the test dataset. We picked four objects as unseen objects in our split, they are gelatin box, mustard bottle, pitcher base and power drill, labelled with bold text in Table 1. In terms of detection difficulty, gelating box and mustard bottle are easy, pitcher base is harder, and power drill is hardest. This conclusion is carried out based on the attributes they have compared to all the seen objects, it is also proved later with the detecting score.

After four unseen objects are picked, 31 videos that only contain seen objects are selected to be used in the train and validation dataset. The rest 61 videos that contain at least one unseen object are allocated to the test dataset. The 31 videos only contain seen objects have 45272 frames in total. We randomly picked 20% of them (9040 images) and put them into the validation dataset, the rest 80% (36232 images) are placed in the train dataset. For test dataset, all frames in the 61 videos (88664 images) are used. In Table 1, we also show the number of labels for each object in each dataset.

Table 1: Number of images and labels for each object in each set

| Number | Object name | Train labels 36232 images | Validation labels 9040 images | Test labels 88664 images |
|---|---|---|---|---|
| 0 | master chef can | 8919 | 2252 | 18088 |
| 1 | cracker box | 10073 | 2552 | 19771 |
| 2 | sugar box | 12393 | 3055 | 16965 |
| 3 | tomato soup can | 13342 | 3213 | 23206 |
| **4** | **mustard bottle** | 0 | 0 | 32321 |
| 5 | tuna fish can | 7534 | 1868 | 21234 |
| 6 | pudding box | 5214 | 1384 | 26334 |
| **7** | **gelatin box** | 0 | 0 | 33786 |
| 8 | potted meat can | 10247 | 2549 | 19354 |
| 9 | banana | 10060 | 2493 | 20275 |
| **10** | **pitcher base** | 0 | 0 | 26478 |
| 11 | bleach cleanser | 11918 | 2933 | 15755 |
| 12 | bowl | 7808 | 1945 | 4898 |
| 13 | mug | 11051 | 2797 | 12858 |
| **14** | **power drill** | 0 | 0 | 27883 |
| 15 | wood block | 6649 | 1669 | 12782 |
| 16 | scissors | 12463 | 3210 | 11710 |
| 17 | large marker | 9211 | 2277 | 19305 |
| 18 | large clamp | 7342 | 1804 | 16661 |
| 19 | extra large clamp | 12841 | 3186 | 9575 |
| 20 | foam brick | 7268 | 1745 | 19413 |

### 4.2 TESTING RESULT

During testing, all images in the test dataset were used. Since this work focus on detection only, we will only evaluate the recall rate of the algorithm. We define that if an object's ground truth bounding

box (GT) and the predicted bounding box's IoU is bigger than 0.5, the object is successfully detected, noted as True Positive (TP). The recall rate is defined as:

$$Recall = \frac{TP}{GT}$$

In the following table, the recall for each object is shown.

Table 2: Recall for each object

| Number | 0 | 1 | 2 | 3 | **4** | 5 | 6 | **7** | 8 | 9 | **10** | 11 | 12 | 13 | **14** | 15 | 16 | 17 | 18 | 19 | 20 |
|---|---|---|---|---|---|---|---|---|---|---|---|---|---|---|---|---|---|---|---|---|---|
| Recall | 0.88 | 0.72 | 0.83 | 0.85 | 0.73 | 0.75 | 0.61 | 0.46 | 0.86 | 0.73 | 0.05 | 0.51 | 0.63 | 0.90 | 0.07 | 0.21 | 0.44 | 0.26 | 0.42 | 0.21 | 0.72 |

### 4.3 DISCUSSION

Base on the testing result, we can say that the algorithm works well when there are similar seen objects, but not for onjects that very different from seen objects. From Table 2, the first thing we notice is that the recall rates for different objects have a large variation. Some seen objects even have a lower recall rate than unseen objects. The main reason causing this is the unbalanced number of labels in the train dataset and the test dataset. For example, the number of train labels for sugar box is 0.75 times of the number of test labels, and its recall reached 0.83. For wood block, its number of train labels is only half the number of test labels, and its recall rate is only 0.21. Another reason that affects the recall rate is the variation of illumination. Since YCB Video dataset is consists of videos, images in the train dataset can only cover a very limited range of illumination conditions. Thus, the algorithm will perform worse on the test images with illumination conditions that have never been met before.

For the four unseen objects, the recall rate for object number 4 and 7 is higher than object number 10 and 14, which is similar to what we expected. Especially for object numer 4, its recall rate is even higher than many seen objects. Object number 4 has the highest recall rate among all unseen objects because its color and shape have commonly appeared on seen objects. However, attributes contained in object number 10 and 14 are hardly seen in the train dataset. Thus, we can conclude that seen objects with similar color or shape to unseen objects can increase the detection rate of unseen objects.

## 5 CONCLUSION

In this paper, we proposed to use a modified YOLOv5 neural network to perform generalized zero-shot detection on seen and unseen objects. We also proposed a novel splitting method for YCB Video dataset to train and test gZSD algorithms. By changing the final detection layers of YOLOv5, we have significantly improved its gZSD performance on the YCB Video dataset split with our proposal. For industrial robots that works in a flexible and dynamic manufacturing environment, our gZSD algorithm for detecting daily objects is a more feasible solution than the traditional vision algorithm that requires training for every object. In our experiment, we found that our algorithm is more sensitive to color rather than shapes. Thus, in the future, we can experiment on RGB-D images rather than RGB images to evaluate the improvement brought by the extra depth channel.

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
