# OpenReview forum: "Zero-shot detection of daily objects in YCB video dataset"
_ICLR.cc/2022/Conference — ICLR 2022 Submitted_

### Official Review · Reviewer_Ajjn · 2021-10-31

**Correctness:** 2
**Technical Novelty And Significance:** 1
**Empirical Novelty And Significance:** 1
**Recommendation:** 1
**Confidence:** 5

**Main Review:**

Strength
1. The paper tackles on a practically very important problem. The problem is well-motivated from the viewpoint of a practical application.

Weakness
1. The technical novelty of the proposed method is very limited. Basically, the difference is only in the attribute prediction part, but the modification is very straightforward even in that part.
1. The experiment is not enough for supporting the effectiveness of the proposed method. It is not compared with any existing methods. There is little analysis.
1. The second contribution claimed by the authors is weak.\
`2.A novel splitting method for YCB Video dataset that splits the dataset by seen and unseen objects.`\
 Actually, they just provide one of the arbitrary splits, and this cannot be regarded as contribution.
1. The paper claims that they tackle on generalized zero-shot detection, but I believe “localization” is more appropriate word than “detection” since the proposed method does not recognize object classes.
1. The manuscript needs careful proof-reading for English style and grammar issues. For example, but not limited to
    - robots will be allocated to various tasks depend on demand. Which requires the robots
to be able to…
    - For every object, their class labeling is translated into attribute vectors that’s contains the colour and shape information of each object.
    - Tensor $T_f$ represents the block of are extracted features in three levels, they are passed to the following blocks
    - It needs to be notice that…
    - our attribute vector is carried out by human eye evaluation
    - For example, object ‘red cup’ has the attributes of red, cup, circle, cylinder will have the attribute vector [0 0 1 0 0 0 0 0 1 0 0 0 1 1 0 0].
    - The 31 videos only contain seen objects have 45272 frames in total
    - YCB Video dataset is consists of …



**Summary Of The Paper:**

This paper tackles on a generalized zero-shot localization task. YOLOv5 network is slightly modified to detect pre-defined attributes. The proposed method is evaluated on YCB video dataset.

**Summary Of The Review:**

I found the paper deals with a practically very important and motivation of the paper is good. However, the novelty of the paper is very limited, and the experiment is not enough for showing the advantage of the proposed method over prior works.

---

### Official Review · Reviewer_uZSA · 2021-10-31

**Correctness:** 1
**Technical Novelty And Significance:** 1
**Empirical Novelty And Significance:** 1
**Recommendation:** 1
**Confidence:** 5

**Main Review:**

Strengths:
1.	The network is designed to output the object attribute vectors, which could be useful when the object set is fixed or strictly limited to these attributes.

Weaknesses:
1.	The proposed method is a simple modification of the YOLOv5 network. The proposed object attribute prediction branch is questionable as the attribute slots are predefined, fixed and do not generalize to new attributes.
2.	The proposed method does not give object ID but only bounding boxes. Thus the method is reduced to an objectness detector. There are many possible baselines (e.g. pretrained YOLOv5 as objectness detector) but the paper compares to none.
3.	The paper does not mention or compare to relevant previous works like [1] and [2]
4.	The paper does not conduct any ablation study, with the proposed method empirically unverified.
5.	The paper only evaluates one dataset and uses the recall as the sole evaluation metrics, ignoring the harm of false positives.
6.	The results are not good, only on average around 0.3 in recall on unseen objects.

Minor issues (grammar, typos, etc):
1.	“since the knowledge can be learned from each object is very limited” -> “since the knowledge that can be learned from each object is very limited”
2.	“ZSL and gZSL algorithms only need to analysis the categorical information” -> “ZSL and gZSL algorithms only need to analyze the categorical information”
3.	“Covert the class labelling to 16 attributes that represents” -> “Convert the class labelling to 16 attributes that represents”
4.	There are still many grammar/typo issues in the paper. I suggest the authors thoroughly check them, although this is not considered in the evaluation.

[1] P. Ammirato, C.-Y. Fu, M. Shvets, J. Kosecka, and A. C. Berg, “Target Driven Instance Detection,” arXiv:1803.04610 [cs], Oct. 2019, Accessed: May 19, 2020. [Online]. Available: http://arxiv.org/abs/1803.04610
[2] J.-P. Mercier, M. Garon, P. Giguère, and J.-F. Lalonde, “Deep Template-based Object Instance Detection,” arXiv:1911.11822 [cs], Nov. 2020, Accessed: Nov. 17, 2020. [Online]. Available: http://arxiv.org/abs/1911.11822


**Summary Of The Paper:**

This paper tries to tackle the zero-shot object instance detection problem, but without handling object classification, it is essentially a method for objectness detection. The paper uses a modified YOLOv5 network with an object attribute output branch. The object attributes are 16 binary attributes pre-defined and fixed by human evaluation. The method is evaluated on the YCB-Video dataset, where 4 out of 21 objects are held out as unseen objects for zero-shot evaluation. The paper reports unconvincing results, and no comparison with related work or ablation studies.

**Summary Of The Review:**

The proposed method lacks novelty or contribution. The experiments are not complete or convincing. The results do not seem promising. Therefore I recommend rejecting this paper.

---

### Official Review · Reviewer_kFeR · 2021-11-03

**Correctness:** 3
**Technical Novelty And Significance:** 1
**Empirical Novelty And Significance:** 1
**Recommendation:** 1
**Confidence:** 4

**Main Review:**

-Strengths: The paper is well-organized and the problem is also well-defined.
-Weakness:
1) The proposed approach is mainly based on YOLO5 and the novelty is really limited. No new or novel zero shot learning algorithms are proposed to solve this specific problem.
2) There's no explanation about why the proposed model can detect the unseen objects.
3) No ablation study are given to validate the effectiveness of each proposed component. There's only on experiment on YCB Video dataset, and the comparison with other state-of-the-arts zero shot learning methods is missing.


**Summary Of The Paper:**

This paper aims to solve the object detection under the object manipulation scenario. More specifically, it attempts to design a zero-shot object detection algorithm that can detect unseen objects with the knowledge learnt from seen objects. The authors conduct experiments on YCB Video dataset to valid the effectivness of the proposed method.

**Summary Of The Review:**

Based on the weakness of this paper mentioned before, I think the quality of this paper is quite low and cannot be accepted.

---

### Official Review · Reviewer_8vns · 2021-11-03

**Correctness:** 2
**Technical Novelty And Significance:** 1
**Empirical Novelty And Significance:** 1
**Recommendation:** 3
**Confidence:** 4

**Main Review:**

The paper presents an interesting application of ZSL. While in general ZSL has been a fairly academic problem, the author's method is able to be used in (constrained) real-world settings with multiple objects within the scene. They pose it in robotics and smart manufacturing however, can be generalised.

The approach to extend Yolo to include the attribute makes a lot of sense, however, it raises the question is all you need is the attribute encoding? No additional contrastive learning, hard-mining. In the results/discussion, you comment on the distribution, this would be a large limitation of the method as it isn't possible to have balanced distributions from real-world images of multiple objects

Evaluation is the most weakest part. Given the relatively simple adaption, different backbones could have easily been evaluated R-CNN ... with their varying backbones. Given the authors are trying to motivate a new problem, only showing their own results is not convincing.

Also, a factor for the evaluation here would be the computation time with the standard time vs accuracy debate in object detection methods.



In the related work the claim "The varieties of objects in one category can increase the generality of the trained algorithm.", is there evidence behind this (needs citation)? In general, this seems counterintuitive. The more general the class the harder it is to classify as key pixels for decision making are harder to select in a broad setting.


Typos / Clarifcations:
Abstract: "objects in a greater field of view rather than being limited to a small space where the object could appear." isn't clear the use of field of view that doesn't seem to make sense in this context
Related work: "many manufacturing planets have replaced" planets = plants
Related work: "They are existing works in" They = There


**Summary Of The Paper:**

The paper tries to generalise the generalised zero-shot learning (gZSL) problem to full images as an object detection problem instead of an image classification problem. They propose the use of the YoloV5 network as a backbone for the prediction of the attribute encoding of the detected bounding boxes. In such a way they can then use the attributes to suppress background objects/predictions based on the overlap of seen and unseen objects. Therefore they combine the localisation and objections loss with an attribute loss (BCE) to train the network. They evaluate over YCB video dataset on 21 objects for their method only.

**Summary Of The Review:**

While the paper presents an interesting application, the method is a very baseline method with little adaption to the prior architecture. The evaluation is missing a benchmark to make this paper competitive and establish the application problem,

It is also a very application problem and may not be the best fit for ICLR but instead a robotics or computer vision conference as it adapts an ML method to a generalised problem.

---

### Official Review · Reviewer_TYzZ · 2021-11-04

**Correctness:** 3
**Technical Novelty And Significance:** 1
**Empirical Novelty And Significance:** 2
**Recommendation:** 3
**Confidence:** 4

**Main Review:**

Strengths

1) The overall problem seems to be practically relevant. Most existing object detectors and classifiers are desired not to overfit to the color or size of the object as long as the objects are from the same class. If the neural network saw only black dogs during training, it is desirable to predict white dogs also in deployment.

2) Generalized zero-shot learning is less explored compared to zero-shot learning.

Weaknesses

1) The novelty is not very strong. Attribute-based zero-shot learning is a common paradigm and is discussed in [A,B,C,D]. Attribute-based ZSD is also discussed in [E]. The authors extend it to detection on a dataset with different classes of objects than seen in other papers. Further, they use YOLOv5 instead of YOLOv2 as the backbone. There are some changes to the neural net architecture but there is no empirical evidence to suggest those are beneficial.

2) The empirical results are underwhelming. The recall ratio is very low for 2 classes. If the authors are claiming zero shot detection, then the method should be able to detect unseen objects ideally even if those attributes are not a lot in the training set. Further, just a single dataset is used and there is no comparison to prior art. If none of the prior art is applicable to this problem, the authors need to explain why that is the case.

3) The attribute list is too limiting and will not scale. These attribute lists are just directed towards this single dataset. For general applications, it might be difficult to list out and label all attributes in all images. Further, a hierarchical label might be more useful. Due to this reason, it might be more beneficial to learn embeddings/representations that are more general though not interpretable.

Questions

1) Why is the mAP which is the standard score for object detection, not reported? The algorithm can still generate false positive.
In [E], the authors report mAP over all classes as they do not classify the objects.

2) More papers on ZSD that are not cited: [F,G,H,I].


References:

[A] https://ieeexplore.ieee.org/document/6571196
[B] https://ieeexplore.ieee.org/document/6126373
[C] https://ieeexplore.ieee.org/document/6751374
[D] https://link.springer.com/chapter/10.1007/978-3-642-15555-0_10
[E] https://arxiv.org/pdf/1803.07113.pdf
[F] https://www.sciencedirect.com/science/article/pii/S2667241321000124
[G]https://openaccess.thecvf.com/content_CVPR_2020/html/Zhu_Dont_Even_Look_Once_Synthesizing_Features_for_Zero-Shot_Detection_CVPR_2020_paper.html
[H] https://ojs.aaai.org/index.php/AAAI/article/view/6868
[I] https://arxiv.org/abs/1805.06157

**Summary Of The Paper:**

The paper performs zero shot detection of seen and unseen objects in scenarios with more fine-grained division of objects. For example, in practical computer vision applications such as industrial and indoor environments, it might be necessary to differentiate between the same object in different colors, sizes and shapes. In this paper, unseen objects can be classified based on the attributes: white, blue, red, yellow, silver, black, brown, bottle, cup, can, clamp, slim, circle, cylinder, box, rectangular. The authors perform zero-shot detection using a simple approach of predicting attributes. The attributes are labeled in the training images and the method has to generalize for unseen objects in the test images.

**Summary Of The Review:**

The problem the paper seeks to solve is relevant and interesting. However, there is insufficient empirical or logical arguments put forth to justify why this approach is good or better than previous approaches. Empirical results are not convincing, the novelty is mainly in the use of a new dataset and literature survey is not thorough.

---

### Decision · Program_Chairs · 2022-01-20

**Decision:**

Reject

**Comment:**

All five reviewers unanimously agree that the paper needs to be rejected. One of the main concerns is the lack of technical novelty/originality. The reviewers also point out lacking citation and comparison to prior work, and missing experiments. The authors have not provided any rebuttal.This paper describes an approach for zero shot detection of seen and unseen objects in scenarios. All five reviewers unanimously agree that the paper needs to be rejected. One of the main concerns is the lack of technical novelty/originality. The reviewers also point out lacking citation and comparison to prior work, and underwhelming experiments. The authors have not provided any rebuttal.

We recommend rejecting the paper.